# Acoustomicrofluidic assembly of oriented and simultaneously activated metal–organic frameworks

Heba Ahmed[1], Amgad R. Rezk[1], Joseph J. Richardson [2,3], Lauren K. Macreadie[4], Ravichandar Babarao[4,5], Edwin L.H. Mayes [5], Lillian Lee [1] & Leslie Y. Yeo [1]

The high surface area and porosity, and limitless compound and network combinations between the metal ions and organic ligands making up metal–organic frameworks (MOFs) offer tremendous opportunities for their use in many applications. While numerous methods have been proposed for the synthesis of MOF powders, it is often difficult to obtain oriented crystals with these techniques. Further, the need for additional post-synthesis steps to activate the crystals and release them from the substrate presents a considerable production challenge. Here, we report an acoustically-driven microcentrifugation platform that facilitates fast convective solutal transport, allowing the synthesis of MOF crystals in as short as five minutes. The crystals are not only oriented due to long-range out-of-plane superlattice ordering aided by molecular dipole polarization under the acoustoelectric coupling, but also simultaneously activated during the synthesis process.

[1] Micro/Nanophysics Research Laboratory, School of Engineering, RMIT University, Melbourne, VIC 3000, Australia. [2] ARC Centre of Excellence in Convergent Bio-Nano Science & Technology, The University of Melbourne, Parkville, VIC 3010, Australia. [3] Department of Chemical & Biomolecular Engineering, The University of Melbourne, Parkville, VIC 3010, Australia. [4] Commonwealth Scientific and Industrial Research Organisation (CSIRO), Normanby Road, Clayton, VIC 3168, Australia. [5] School of Science, RMIT University, Melbourne, VIC 3000, Australia. Correspondence and requests for materials should be addressed to L.Y.Y. (email: leslie.yeo@rmit.edu.au)

Metal–organic frameworks (MOFs)—highly ordered three-dimensional coordination networks comprising inorganic nodal units interconnected by polytopic organic ligands—have recently garnered significant attention because of their exceptionally high Brunauer–Emmett–Teller (BET) surface areas (~$10^4$ m²/g) and porosities (up to 90% of its free volume)[1,2]. Moreover, their structural diversity, arising from the vast number of possible combinations between the metal nodes and organic linkers, facilitates the tailoring of materials with widely different physical, chemical, and geometrical properties for applications that span catalysis, gas separation and sensing, charge transport and storage, and drug delivery, amongst others[2–8].

While MOFs have conventionally been synthesized through a variety of techniques, including hydrothermal, solvothermal, microwave, sonochemical, and electrochemical synthesis[9], drawbacks associated with these routes include the random orientation, polycrystallinity and defect-rich nature of the MOFs produced due to inhomogeneities in the diffusion process[10,11]. Moreover, a significant practical limitation in the large-scale production of these MOF powders is the requirement for subsequent post-synthesis chemical or thermal activation to remove the unreacted solvents trapped within the pores[12–14]. Furthermore, in some instances, thermal and chemical activation via solvent exchange hinder the use of MOFs as green materials, and has often failed to yield the expected internal surface area.

In this work, we report an acoustomicrofluidic platform that facilitates one-step synthesis and activation of MOF powders with high degree of orientation and surface area. The facile, room-temperature operation is rapid—MOFs are produced in as little as 5 min, and are found to already be activated during their synthesis, therefore removing the need for further post-processing activation steps. In particular, we demonstrate the synthesis of HKUST-1 ($Cu_3(1,3,5$-benzenetricarboxylate)$_n$; Cu–BTC) since it is a particularly well-characterized MOF[15,16] that facilitates comparison of its geometrical properties. In addition, we show that the technique can also be extended with similar results for the synthesis of Fe-MIL-88B [$Fe_3O(1,4$-benzenedicarboxylate)$_3$Cl]. That the HKUST-1 and Fe-MIL-88B crystals are simultaneously activated during their synthesis in the current process is significant, not just from the standpoint of reducing processing time, complexity and failure, but also from an economic and environmental cost perspective since conventional activation methods under vacuum or via liquid solvent exchange are either energy-intensive or results in waste organic solvents that necessitate facilities for their treatment or disposal, particularly in large-scale manufacture.

## Results

**Synthesis and activation of HKUST-1.** The acoustomicrofluidic synthesis platform, which comprises a piezoelectric substrate (lithium niobate; $LiNbO_3$), is schematically shown in Fig. 1a. The pair of interdigital transducers (IDTs) are deliberately patterned off-center on the substrate in order to break the symmetry of the opposing surface acoustic waves (SAWs)—nanometer-amplitude MHz-order electromechanical Rayleigh waves (longitudinal, transversely-polarized, i.e., out-of-plane, surface-propagating

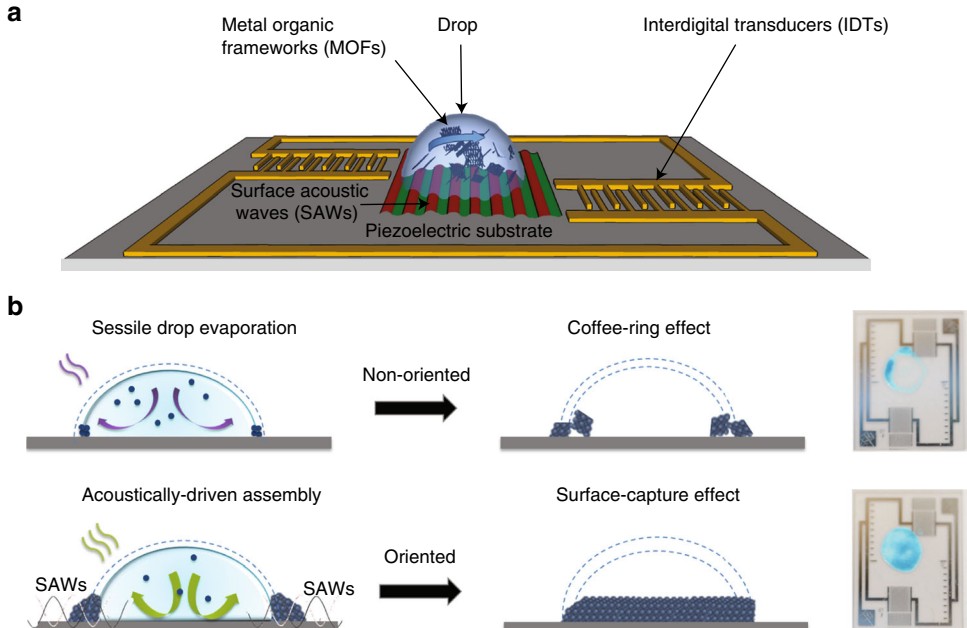

**Fig. 1** Rapid synthesis of oriented and activated MOFs. **a** Illustration of the acoustomicrofluidic platform on which the MOF crystals are synthesized. Opposing SAWs are generated on a piezoelectric substrate by applying an input voltage to a pair of offset IDTs patterned on the substrate. Coupling the asymmetric SAW energy into a ml sessile drop containing the precursor solutions then produces a microcentrifugation flow, which drives the subsequent precipitation and nucleation of the MOF crystals within it. **b** Schematic depiction of the postulated mechanism by which the MOFs are synthesized. The top row illustrates the control experiment in the absence of the SAW irradiation in which slow solvent evaporation leads to a weak convection cell in the drop, which transports the solute molecules to its contact line, where they precipitate to form a ring of crystals akin to coffee-ring stains. The slow diffusion-dominated process then culminates in a dilute solute concentration in the contact line region, and therefore the crystals that form lack long-range ordering. The bottom row depicts the case under Rayleigh SAW excitation, wherein the microcentrifugation flow that arises drives fast turbulent convective transport of the solute molecules to the contact line, whose oscillation smears out the coffee-rings, leading to homogeneous deposition of successive stacks of solute monolayers within this highly concentrated region. Aided by the evanescent electric field from the SAW, this results in vertical-oriented stacking of the monolayers, culminating in a large, ordered superlattice structure, thus explaining why oriented MOF crystals are obtained under the acoustic excitation

compressional waves)—they generate upon application of an oscillating electric field at resonance. The transmission of these asymmetrically opposing waves into a 10 μl sessile liquid drop placed atop the substrate then results in an internal microcentrifugal flow[17–21] that has been previously demonstrated for driving extremely efficient micromixing and particle concentration[22]. Subjecting a drop containing 5 μl of a copper precursor, i.e., copper(II) nitrate hemi(pentahydrate) (Cu(NO$_3$)$_2$ · 2.5H$_2$O), and 5 μl of trimesic acid (benzene-1,3,5-tricarboxylic acid; H$_3$BTC), both in 1:1 (vol/vol) ethanol–water solutions, to such acoustically-driven microcentrifugation at varying acoustic intensities (1.5, 4.5, 7.5, and 9 V$_{rms}$) for 5 min can be seen to induce nucleation and subsequent crystallization of HKUST-1 (Fig. 2), which is amongst the foremost and most common of MOFs reported in the literature given its excellent thermal stability, and superior adsorption and catalytic properties[23,24].

Confirmation of the production of stable HKUST-1 MOFs is provided by the Fourier Transform Infrared (FTIR) spectra in Supplementary Fig. 1 in which we verify the asymmetric stretching of the carboxylate groups in the H$_3$BTC molecules at 1508–1623 cm$^{-1}$ and the symmetric stretching of the COO–Cu$_2$ carboxylate groups at 1384 and 1405 cm$^{-1}$. Several bands over wavenumbers 1300–600 cm$^{-1}$ are observed, which can be attributed to the out-of-plane vibration of the H$_3$BTC molecules[25,26]. Noting the thermal stability of solid HKUST-1 to exceed 300 °C, thermal gravimetric analysis (TGA) of one of the samples (9 V$_{rms}$), on the other hand, reveals two major stages in the weight loss behavior of the HKUST-1 crystals, consistent with that observed for bulk HKUST-1 (Supplementary Fig. 2)[27]. The first weight loss stage occurs at temperatures below 200 °C, which can be attributed to the removal of water and solvent molecules from the surfaces of the HKUST-1 precursors. For octahedral HKUST-1 crystals, the second weight loss stage starts at around 270 °C and ends around 340 °C. The relative weight loss (weight loss/residual weight × 100%) of the HKUST-1 crystals at this stage is approximately 36.618%, roughly corresponding to the theoretical weight loss caused by the combustion of the organic ligands in HKUST-1 in air. With increasing temperature, we observe a slight weight decrease of approximately 8.866% in the temperature range between 330 °C and 450 °C, which can be attributed to the oxidation of Cu$_2$O in air (Cu$_2$O + 1/2O$_2$ → 2CuO).

Further inspection of the orientation of the resultant crystals reveals its strong dependence on the magnitude of the acoustic energy coupled into the drop, which is accompanied by an intensification of the convective microcentrifugation flow (Fig. 2). We first observe from the scanning electron microscopy (SEM) images and size distributions in Fig. 2 (left and center columns) that increasing the flow intensity results in octahedral crystals typical of HKUST-1 that are progressively smaller and more homogeneous in size, decreasing from a mean characteristic dimension of 53.49 ± 4.04 μm at 1.5 V$_{rms}$ to 34.19 ± 4.95 μm at 4.5 V$_{rms}$, 25.83 ± 1.37 μm at 7.5 V$_{rms}$ and 15.09 ± 1.78 μm at 9 V$_{rms}$, as compared to that of bulk HKUST-1 with a mean diameter of 73.15 ± 5.85 μm (Supplementary Table 1). This dimensional reduction is a consequence of the turbulence generated in the drop due to the leakage of the SAW energy into it, given streaming Reynolds numbers[28] Re$_s$ ≡ $\hat{u}\mathcal{L}/\nu \approx 10^3$ well above the $10^2$ critical value reported for the classical transition to subharmonic turbulence in acoustical flows[29,30]; here, $\nu$ denotes the kinematic viscosity of the fluid, $\hat{u}$ its Lagrangian velocity and $\mathcal{L}$ the characteristic drop dimension. The turbulent mixing eddies generated in the flow then results in enhanced convective transport, which is known to lead to the formation of smaller crystals[31,32], given that the eddy

size imposes an upper limitation to the crystal dimension during its growth. This is confirmed not only by the order of magnitude agreement between the Kolmogorov length scale[33] $\eta = (\nu^3\mathcal{L}/\mathcal{U}^3)^{1/4} \sim \mathcal{O}(10\,\mu m)$, characteristic of the eddy size with the crystal dimension observed, but also the $\mathcal{U}^{-3/4}$ velocity scaling that the crystal dimension obeys, as seen in Supplementary Fig. 3.

More significantly, we note from the powder X-ray diffraction (XRD) spectra in the right column of Fig. 2b–e that the HKUST-1 crystals synthesized from the present technique exhibit a high degree of orientation parallel to the {222} plane, especially at high input voltages. This is in stark contrast to the control experiment in which the crystals that form under slow solvent evaporation of the same drop on identical substrates in the absence of the acoustic forcing show no apparent orientational preference (Fig. 2a, right column). Interestingly, we observe that the increase in the input voltage leads to more prominent vertical, out-of-plane orientation, as can be seen by the appearance of additional peaks parallel to the {222} plane, such as the {333}, {444} and {555} planes at 2θ = 11.7°, 17.7°, 23.7°, and 29.7°, respectively (Fig. 2(b–e), right column).

We note in the XRD spectra a slight shift in the peaks (≈ + 0.4°), which can be attributed to the effect of compressive stresses on the crystal lattice structure arising from the acoustic forcing; a similar observation was recently reported for sodium chloride crystals[34]. To obtain different deformed models of Cu–BTC, density functional theory (DFT) simulations were carried out using a volume-conserving strain tensor applied to the lattice parameters with strain magnitudes ranging from −0.008 to 0.008 in 0.002 increments. A comparison of the theoretical and experimental powder XRD data using the Pawley fitting method[35] then yielded refined cell parameters which accurately resemble that expected for HKUST-1. Matching of the data for the crystals acquired under input voltages of 1.5 and 4.5 V$_{rms}$ was obtained when a strain of −0.002 that led to slight compression along the c-plane was applied, resulting in cell parameters a = b = 26.30597 Å and c = 26.62462 Å with α = β = 90° and γ = 89.65623°. For the crystals acquired when the input voltage was increased to 7.5 V$_{rms}$, matching was obtained with a slight increase in the applied tensor strain of −0.006 that led to more compression along the c-plane to yield cell parameters a = b = 26.30597 Å and c = 26.62483 Å at α = β = 90° and γ = 89.88541°, whereas further compression by applying a strain of −0.008 along the c-plane was required for the crystals obtained when the input voltage was ramped to 9 V$_{rms}$, yielding cell parameters of a = b = 26.30597 Å and c = 26.62502 Å at α = β = γ = 90°.

Additionally, we note the full-width at half maximum (FWHM) value of the peak at 2θ = 11.7° was seen to decrease with increasing acoustic intensity, from 0.14° at the lowest input voltage (1.5 V$_{rms}$), corresponding to a crystallite size of 59.7 nm, to 0.49° at 9 V$_{rms}$, corresponding to a crystallite size of 167.17 nm. This increase in crystallite dimension with increasing electro-acoustic coupling into the fluid provides further confirmation of the increasing degree of orientation of the crystals. We also note that the crystallite size at 9 V$_{rms}$ is larger than the typical 100 nm sizes observed for HKUST-1 surface anchored MOFs (SUR-MOFs) synthesized using the layer-by-layer technique after 80 cycles—a much longer process requiring several hours[36,37].

A possible mechanism by which the out-of-plane crystal orientation arises can be postulated from the solutal dynamics associated with the acoustically-driven microcentrifugation flow. In the absence of the acoustic excitation, the weak convective flow that arises in a sessile drop left to slowly evaporate transports solute molecules to its contact line where the locally singular evaporation rate drives their precipitation to form a particulate

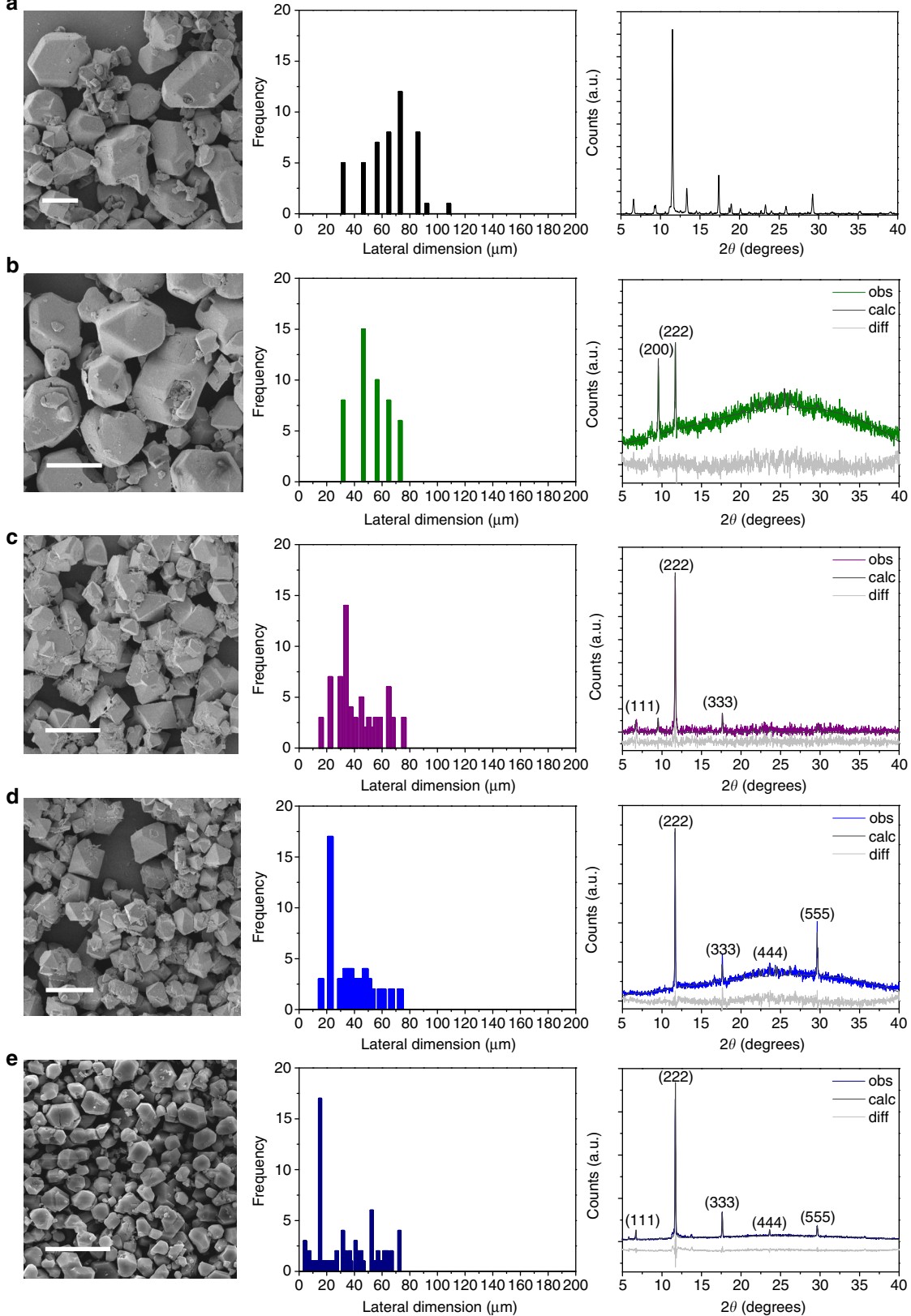

**Fig. 2** Morphology, size and orientation of the HKUST-1 MOFs. SEM scans (left column; scale bars denote lengths of 50 μm), corresponding lateral size frequency distributions (from an analysis of approximately 100 crystals; center column), and, (powder XRD patterns (right column) of **a** bulk HKUST-1 synthesized under slow solvent evaporation as the control in the absence of acoustic excitation, and, **b**–**e** HKUST-1 MOFs synthesized under increasing input voltages (1.5, 4.5, 7.5, 9 $V_{rms}$, respectively). The Pawley fit (calc) to the experimental (obs) powder XRD data is shown in dark gray and the difference (diff) is shown in light gray, from which it can be seen that the observed XRD patterns matches that of crystal models that exhibit preferential orientation

ring—the well-known coffee-ring stain effect[38–41], as illustrated in the top row of Fig. 1b. More importantly, the diffusion-dominated transport, with a timescale on the order $\mathcal{L}^2/\mathcal{D} \sim \mathcal{O}(10^3\,\mathrm{s})$, wherein $\mathcal{D}$ is the molecular diffusion coefficient, is too slow to enhance the local solutal concentration immediately behind the drying front, i.e., the contact line region, such that the weak intermolecular interactions that arise between the solute molecules are insufficient to result in large long-range vertical ordering of the crystal.

On the other hand, the oscillation of the contact line under the MHz-order SAW vibration[42] as it recedes in a stick–slip manner (a consequence of pinning effects in the presence of surface heterogeneities[39]) is observed to smear out the rings[43,44], leading to the successive deposition of monolayers across the entire footprint of the drop (Fig. 1b, bottom row). The turbulent convective flow driven by the acoustics is also sufficiently fast—with time scales on the order $\mathcal{L}/\mathcal{U} \sim \mathcal{O}(1\,\mathrm{s})$—in transporting the solute to a large region behind the drying front. Together with the locally singular evaporation rate at the contact line, this results in an exponential enrichment of the local solutal concentration in that vicinity, which provides ripe conditions that are known to promote the formation of large, ordered three-dimensional superlattice structures[45]. The out-of-plane assembly of successive monolayer stacks is further aided by the leakage of the electric field associated with the SAW on the piezoelectric substrate[46,47]; the ability of an induced electric field in controlling the crystal orientation having been previously reported[48]. As illustrated in the left and center columns of Fig. 3a, this evanescent electric field in the liquid induces dipoles in the solute molecules, whose polarization results in their vertical stacking along the field gradient orthogonal to the substrate surface, similar to that in ref. [48].

The increase of out-of-plane crystal orientation with the input voltage as seen in the right column of Fig. 2b–e is consistent with such a theory. To test this hypothesis, we repeated the experiments using shear-horizontal SAWs (SH-SAWs), i.e., longitudinal, horizontally-polarized, i.e., in-plane, surface propagating shear waves, generated on a lithium tantalate (LiTaO₃) substrate, whose native electric field polarization occurs in the plane of the substrate (Fig. 3b, left and center columns), in contrast to that of the Rayleigh SAW, whose native electric field polarization occurs out of the plane of the substrate (Fig. 3a, left

and center columns). Under the same conditions, we observe the crystals produced to possess an in-plane orientation, i.e., parallel to the {200} plane, as shown in the right column of Fig. 3b. We also note the negligible temperature change (from 27.8 to 28.6 °C) in the drop over the acoustic excitation period, even for the highest input voltage, thus eliminating the possibility of heating effects due to either Rayleigh SAW and SH-SAW irradiation on the evaporation dynamics (see Supplementary Fig. 4).

Additionally, the SAW excitation along the substrate results in oscillations in the MOF crystalline structure, which, in turn, squeezes the solvents out of the pores, leading to their simultaneous activation, as observed by the color change in the crystals from a light (solvent-rich) to deep (solvent-poor) purple shade under increasing acoustic field intensities (Fig. 4a). The surface area of the synthesized MOFs are reported in Fig. 4b, in which the N₂ adsorption isotherm at 77 K for input voltages of 1.5, 4.5, 7.5 and 9 V$_{rms}$ yields BET surface areas of approximately 849, 1300, 1434, and 1682 m² g⁻¹, respectively. The latter values are comparable to that reported in the literature for commercially-available activated HKUST-1 crystals[15] (1500–2100 m² g⁻¹) and more than double the surface area (≈700 m² g⁻¹) associated with the HKUST-1 crystals produced in the absence of SAW excitation, which matches closely with that reported for the same crystals prior to their activation[49].

**Synthesis and activation of Fe-MIL-88B.** Similar oriented and simultaneously-activated structures were also observed for the synthesis of Fe-MIL-88B crystals—which comprises a three-dimensional hexagonal structure built up from trimers of FeO₆ octahedra linked to benzenedicarboxylate anions forming water-filled tunnels along the c-axis connected by bipyramidal cages, and thus having relatively small pore dimensions. MIL-88B is especially useful particularly for selective adsorption of gases and solvents because its unit cell is able to shrink and swell reversibly under external stimuli or upon solvent removal during activation[50–52]. We, however, note that the oriented and simultaneously-activated structures were only obtained for exposure to the acoustic irradiation at the highest input voltage of 9 V$_{rms}$ over the same five minute period. The rod-shaped crystals with average lengths of 2–3 μm (Fig. 5a) were seen to be oriented

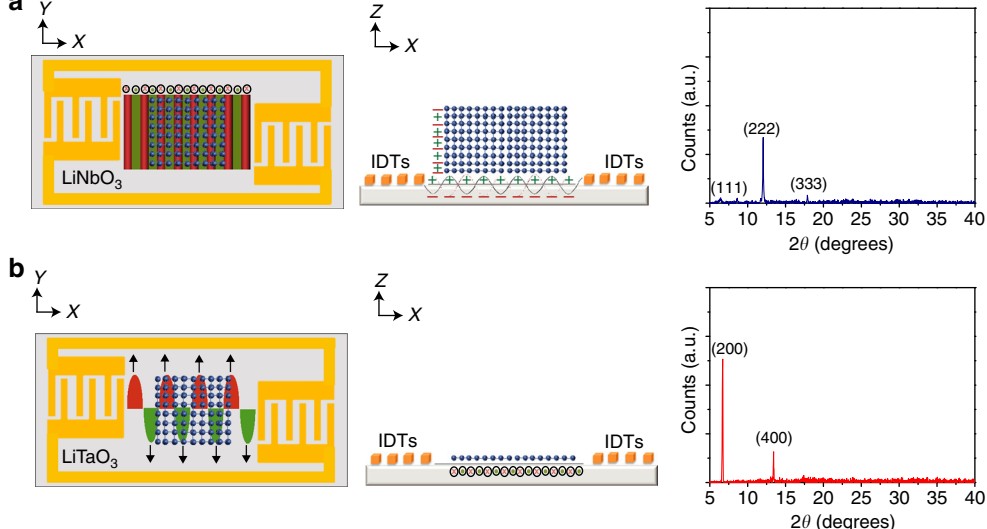

**Fig. 3** Orientation of the HKUST-1 MOFs under acoustoelectric excitation. Top (left column) and side (center column) view schematics (not to scale) illustrating the mechanism by which the HKUST-1 crystals are oriented (or not), as evidenced by the powder XRD spectra (right column) for the case of **a** Rayleigh SAW and **b** SH-SAW excitation at 4.5 V$_{rms}$

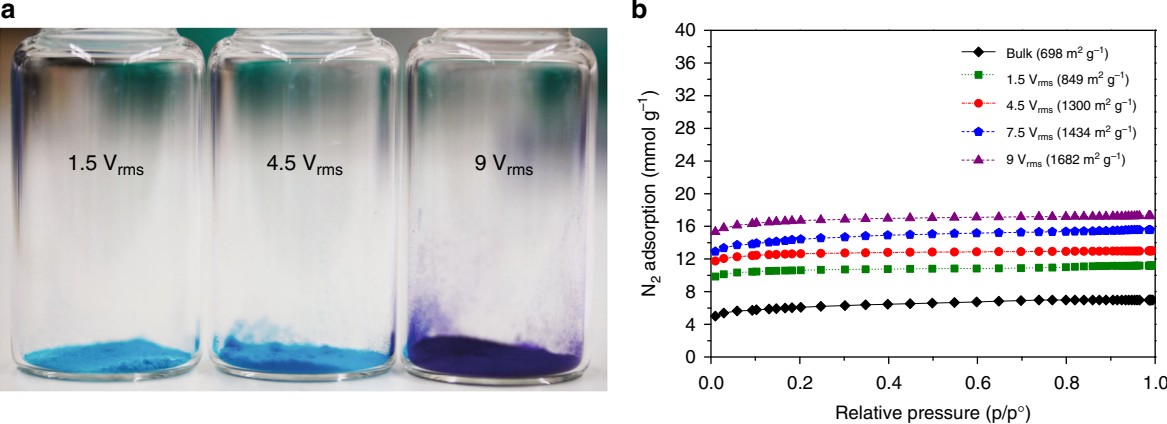

**Fig. 4** Activation and surface area of the HKUST-1 MOFs. **a** The HKUST-1 MOFs that were synthesized under the Rayleigh SAW excitation can be seen to vary in color from a solvent-rich light blue shade to a solvent-poor dark blue shade depending on the input voltage that is coupled into the drop, indicating that the crystals are progressively activated simultaneously during the synthesis as the intensity of the acoustic energy into the drop is increased. **b** $N_2$ sorption isotherms for the HKUST-1 MOFs synthesized at 1.5 $V_{rms}$ (green dashed line), 4.5 $V_{rms}$ (red dashed line), 7.5 $V_{rms}$ (blue dashed line) and 9 $V_{rms}$ (purple dashed line) compared to that for bulk HKUST-1 (black solid line)

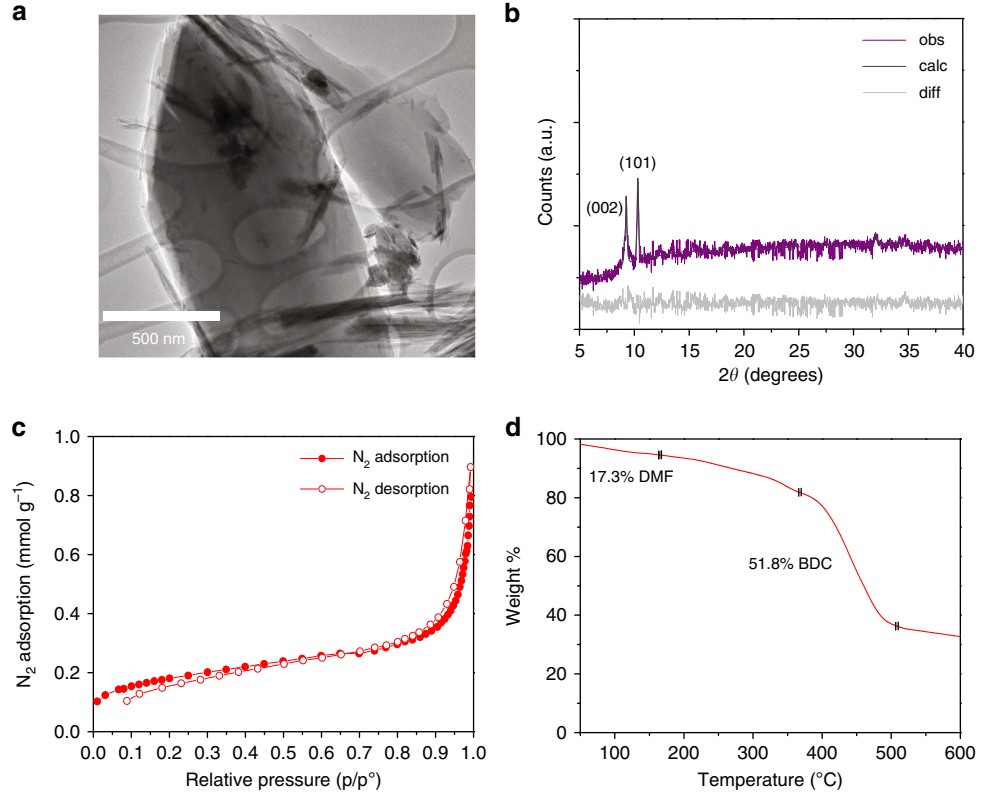

**Fig. 5** Morphology, orientation, surface area and thermal profile of the Fe-MIL-88B MOFs. **a** Transmission electron microscope image, **b** powder XRD pattern (obs; purple) against that simulated from the Cambridge Structural Database[53,67] (CCDC-1485530; calc; dark gray) with differences (diff) shown in light gray, **c** $N_2$ sorption isotherm showing a BET surface area of 8.69 $m^2 g^{-1}$ and, **d** thermal gravimetric analysis (TGA) curves for Fe-MIL-88B MOFs synthesized at an input voltage of 9 $V_{rms}$. The latter shows an approximate 17% weight loss of surface adsorbed DMF solvent below 200 °C, followed by a further 52% weight loss due to decomposition of the 1,4-benzenedicarboxylic acid (BDC) organic linker above 300 °C

towards the {002} plane, together with the appearance of the accompanying {101} plane (Fig. 5b), characteristic of oriented MIL-88B diffraction reported in the literature[52–54]. BET analysis of these MOFs (Fig. 5c) revealed no accessible porosity for $N_2$ at 77 K due to the large contraction of the pores after dehydration; the BET surface area obtained (≈8.7 $m^2 g^{-1}$) being consistent with that reported previously for closed activated MIL-88B

structures[55,56]. Parenthetically, we note that the activation of mesoporous MOFs with even larger pores sizes than MIL-88B, such as MIL-101, using high boiling point solvents can be particularly challenging due to the tendency of the pores to collapse upon removal of the solvent, or the propensity for pore blockage as a consequence of partial solvent removal; both these challenges can be circumvented in the present case simply by adjusting the

input power and duration that the MOFs are exposed to during acoustic activation[14]. Thermal behavior of the samples, on the other hand, shows free and/or coordinated solvent/water departure occurring up to 200 °C, whereas the MIL-88B exhibits thermal stability up to around 300 °C, typical of these classes of MOFs (Fig. 5d)[57].

## Discussion

In summary, we have demonstrated that oriented and simultaneously-activated MOF crystalline powders can be synthesized at room temperature in as little as five minutes (comparable to other synthesis techniques, such as microwave assisted synthesis[56,58]) using an acoustomicrofluidic platform. In addition to its simplicity and speed, as well as the advantages of simultaneous synthesis and activation in a single step, the process easily affords fine control over the out-of-plane architecture of the crystals through the power intensity delivered to the device. Given the low cost of the device (typically around US$1 each) and a production rate of 2 g h$^{-1}$ device$^{-1}$, we note the potential of the setup to be scaled to achieve industrially-relevant production rates by employing a large number of devices in parallel given an equivalent space-time yield $\sigma$P of 17,500 kg m$^{-3}$ day$^{-1}$, thus making the platform an attractive alternative for environmentally-friendly large-scale MOF production for a vast array of applications[59].

## Methods

**Device fabrication**. The acoustomicrofluidic device shown in Fig. 1a consists of a piezoelectric substrate that comprises either 127.68° $Y$–$X$ lithium niobate (LiNbO$_3$; Roditi Ltd., London, UK) for the Rayleigh SAW experiments or, 42° $Y$–$X$ lithium tantalate (LiTaO$_3$; Fujitsu Laboratories Ltd., Atsugi, Japan) for the SH-SAW experiments, on which an off-centered pair of 300 nm thick straight aluminium interdigitated transducers (IDTs) in a basic full-width interleaved configuration are patterned atop a 20 nm thick chromium layer using sputter deposition and standard UV photolithography. The substrates were optically polished on both sides to render it transparent such that the interior of the fluid drop can be observed from the underside of the device to avoid optical distortion at the liquid–air interface of the drop when visualizing from above. Each IDT consists of 25 finger pairs with an aperture of 12 mm and a gap and width of 110 μm, such that application of a sinusoidal electrical signal through an RF signal generator (N9310A; Agilent Technologies, Santa Clara, CA, USA) and amplifier (10W1000C; Amplifier Research, Souderton, PA, USA) at their resonant frequency of 19.37 MHz gives rise to a Rayleigh SAW (in the case of the LiNbO$_3$ substrate) or a SH-SAW (in the case of the LiTaO$_3$ substrate) with a wavelength of 200 μm. A variety of input voltages (1.5, 4.5, 7.5, and 9 V$_{rms}$) to the electrical signal was used in the study, but this was limited to an upper value of 9 V$_{rms}$ to avoid the liquid being nebulized off the device.

**MOF synthesis**. The HKUST-1 working solution was prepared by dissolving the metal precursor, i.e., 0.875 g (3.62 mmol) copper(II) nitrate hemi(pentahydrate) (Cu(NO$_3$)$_2$ · 2.5H$_2$O; Sigma Aldrich Pty. Ltd., Castle-Hill, NSW, Australia), and the organic ligand precursor, i.e., 0.42 g (2 mmol) trimesic acid (C$_6$H$_3$(CO$_2$H)$_3$; Sigma Aldrich Pty. Ltd., Castle-Hill, NSW, Australia), in two separate tubes, each containing 12 ml 1:1 (vol/vol) ethanol (Sigma Aldrich Pty. Ltd., Castle-Hill, NSW, Australia) and MilliQ® water (18.2 MΩ.cm, Merck Millipore, Bayswater, VIC, Australia). A 10 μl drop of this solution was then carefully pipetted onto the middle of the device such that one-half of the drop was subjected to the SAW irradiation in one direction from one IDT and the other half was subjected to the SAW irradiation from the opposite direction from the other IDT. Due to this asymmetry, an azimuthal microcentrifugation flow is generated within the drop (Fig. 1a). After 5 min of exposure to the SAW excitation for each input voltage, during which the MOF was observed to nucleate and hence crystallize, the MOF powder was subsequently collected from the device in an Eppendorf® tube (Eppendorf South Pacific Pty. Ltd., North Ryde, NSW, Australia) and reconstituted with 1:1 (vol/vol) ethanol/water to make a 1 ml solution. This suspension was then centrifuged at 5000 rpm for 5 min and thrice washed in 50% ethanol/water, after which it was left to dry at 25 °C in a sealed glass vial prior to further analysis.

Bulk HKUST-1 was prepared using a hydrothermal synthesis method reported in the literature[60], in which 0.42 g (2 mmol) BTC was dissolved in 24 ml of 1:1 (vol/vol) ethanol/water. The mixture was stirred for 10 min until a clear solution was obtained. Subsequently, 0.875 g (3.62 mmol) of (Cu(NO$_3$)$_2$ · 2.5H$_2$O) was added to the mixture, followed by agitation for a further 10 min. Once the reactants were

completely dissolved in the solvent, the resulting blue solution was left to evaporate to allow the crystallization to occur, through which a blue crystalline powder was obtained. The powder was then washed thrice with 60 ml of a 1:1 (vol/vol) ethanol/water solution and the product left to dry at 25 °C in a sealed glass vial for further analysis.

A similar procedure as that detailed above was repeated for the synthesis of Fe-MIL-88B MOFs, whose working solution was prepared by separately dispersing its precursors, i.e., 31.9 mg iron(III) chloride hexahydrate (FeCl$_3$·6H$_2$O; Alfa Aesar GmbH & Co KG, Lancashire, United Kingdom) and 19.1 mg 1,4-benzenedicarboxylic acid (C$_6$H$_4$(CO$_2$H)$_2$; Sigma Aldrich Pty. Ltd., Castle-Hill, NSW, Australia), in 2.5 ml dimethylformamide (DMF; Thermofisher Scientific, Waltham, MA, USA).

**MOF characterization**. SEM imaging (Philips XL30, FEI, Hillsboro, OR, USA) was employed to characterize the morphology of the MOF crystals. Briefly, the crystals were deposited on a silicon wafer above which a 5 nm gold layer was sputtered over 60 s and imaging was carried out at 10 kV. The size of the MOF crystals was determined through visual inspection of approximately 100 crystals from the SEM digital images using ImageJ (v.1.34, National Institutes of Health, Bethesda, MD, USA). Transmission electron microscopy (TEM; JEM-2100F, JEOL, Frenchs Forest NSW, Australia) images were also obtained by drop casting 3 μl of the MIL-88B crystals onto a holey carbon grid, which was left to dry in air prior to imaging under an acceleration voltage of 80 keV together with a CCD camera (Orius SC1000, Gatan Inc., Pleasanton, CA, USA); post-processing was carried out using the supplied image analysis software (Digital Micrograph® v2.31, Gatan Inc., Pleasanton, CA, USA).

To resolve the crystal structure, powder XRD (D8 Advance, Bruker Pty. Ltd., Preston, VIC, Australia) was conducted with Cu K$\alpha$ radiation at 40 mA and 40 kV ($\lambda$ = 1.54 Å), and a 2$\theta$ range of 6°–50° with a step size of 0.02°; the scan rate was 2° min$^{-1}$ for HKUST-1 and 0.0142° min$^{-1}$ for MIL-88B. The analysis was performed at room temperature and atmospheric pressure. Pawley refinements on the collected powder XRD data were performed using Topas Academic (v4.1.1; Coelho Software, Brisbane, Australia)[61]; this method is suitable for crystalline powders with preferred orientation since the Pawley fitting treats the peak areas as variables[35]. A Gaussian function was used to model the background, in combination with a freely refining Chebyshev polynomial using 8 parameters.

FTIR analysis of the samples at room temperature were acquired using a spectrophotometer (Spectrum One; PerkinElmer Inc., Waltham, MA, USA) by placing a 10 μl suspension of the crystals on a diamond substrate, from which transmittance measurements were conducted in the wavenumber range between 500 and 4000 cm$^{-1}$. The thermal properties of the crystals, on the other hand, were analyzed through TGA (Pyrus 1, PerkinElmer Inc., Waltham, MA, USA). Specifically, 7.5 mg of the crystals were placed in an aluminium stainless steel pan and heated at a rate of 10 °C min$^{-1}$ under N$_2$ from 35 °C to 800 °C. The BET and Langmuir surface areas were calculated from N$_2$ physisorption measurements by placing approximately 0.5 g of the crystals in a surface area and porosity analyzer (ASAP 2020; Micromeritics Instrument Corp. Norcross, GA, USA) under N$_2$ at 77 K, following degassing of the samples at 25 °C under vacuum overnight prior to measurement. To quantify the overall yield and production rate, the dried MOF powder was weighed using a microbalance (XP56; Mettler Toledo Ltd., Port Melbourne, VIC, Australia). Temperature measurements, on the other hand, were carried out using a handheld thermal camera (Trotec EC060V; Emona Instruments, Pty. Ltd., Melbourne, VIC, Australia).

**DFT modelling**. Geometry optimization of perfect and deformed models of Cu–BTC (HKUST-1) were performed using the dispersion corrected density functional theory (DFT-D3) method[62], as implemented in the Vienna Ab Initio Simulation Package (VASP, v5.4.4; VASP Software GmbH, Vienna, Austria)[63]. The initial periodic model for the Cu–BTC structure was taken from previous work[23,64]. Electron exchange and correlation were described using the generalized gradient approximation Perdew, Burke, and Ernzerhof (PBE)[65] form and the projector-augmented wave potentials were used to treat core and valence electrons[66]. In all cases, we used a plane-wave kinetic energy cut-off of 600 eV and a Gamma-point mesh for sampling the Brillouin zone. To obtain different deformed models of Cu–BTC, a volume-conserving strain tensor was applied to the lattice parameters with a strain magnitude that ranged from −0.008 to 0.008 in increments of 0.002. The models were also DFT-D3 optimized, keeping the volume and the lattice parameters fixed while completely relaxing the ionic positions.

## Data availability

The powder XRD, BET and crystal structure data that support the plots within this paper and other findings of this study are available on the Figshare repository (https://doi.org/10.6084/m9.figshare.8020010).

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

## Acknowledgements

The authors acknowledge access to the facilities at the RMIT Microscopy & Micro-analysis Research Facility (RMMF), the RMIT Integrated Victorian X-Ray Structural Determination & Materials Characterisation Facility and the helium ion microscope at the Materials Characterisation and Fabrication platform at the University of Melbourne. They also thank Mr. Frank Antolasic and Ms. Nadia Zakhartchouk for technical assistance. A.R.R. and L.L. are grateful for support through RMIT Vice-Chancellor Post-doctoral Research Fellowships. R.B. acknowledges use of the National Computing Infrastructure (NCI) and the Pawsey supercomputing facilities and funding from the Australian Research Council (ARC) for a DECRA fellowship (DE160100987). LYY is funded through an ARC Future Fellowship (FT130100672) and an ARC Discovery Project (DP170101061).

## Author contributions

H.A., A.R.R., J.J.R., and L.Y.Y. conceived the original research idea. H.A. carried out the device development, sample synthesis and characterization with assistance in data analysis from A.R.R., J.J.R., L.L., and L.Y.Y. J.J.R. and E.L.H.M. conducted the microscopy. R.B. performed the DFT simulations, while L.K.M. carried out Pawley refinement of the data. All authors contributed, discussed, and wrote the manuscript.

## Additional information

**Competing interests:** The authors declare no competing interests.

