## [Peer Review File · Nature Communications]

Reviewers' comments:

Reviewer #1 (Remarks to the Author):

The authors reported a fast way to synthesize HKUST-1 by an acoustically-driven method, as a potential candidate for growing HKUST-1 (and others similar MOFs) rapidly (synthesis times around 5 min) with a great control over crystal size, orientation, and defects. The method described is an alternative to liquid-phase epitaxy pathway, that consists in surface crystallization of materials "layer-by-layer" in a solution in contact to a controlling surface (SURMOFS). The advantages are faster synthesis times and in-situ activation of the solvent molecules in the as-synthesized materials.

The findings are of importance for the MOF community as well as related disciplines (catalysis, gas-adsorption, sensing) as this new approach offers a fast and defined synthesis of MOF materials.

The results and discussion are convincing and the conclusions are clear.

I believe that this paper could lead to a general trend towards this alternative synthesis pathway.

Minor aspects:

The XRD reflections for the acoustically-synthesized materials are shifted (+0.4° approx.). An explanation for this shift should be given. The Miller indexes from the XRD reflections do not correspond to the marked values if you do not consider that shift.

The temperatures given in Fig S4 do not correspond to the commented values in text. The temperature difference is not the same. The authors should clarify which what temperature they were using.

Reviewer #2 (Remarks to the Author):

The manuscript comes from Prof. Yeo and colleagues reported acoustomicrofluidic assembly of MOFs. At the first glance, I was fascinated by the idea that the free-standing MOF film can be highly-oriented and simultaneously-activated. We all know that the highly-oriented/monocrystalline thin film of MOFs is essential for fabrication of MOF-based devices and is really challenging except the layer-by-layer growth. Furthermore, activation of many MOFs is really time-consuming and make many drawbacks in improve the performance of MOF-based devices. If this manuscript can address the two problems by acoustomicrofluidic assembly, it is definitely a big step towards practical application of MOFs.

However, after evaluation of this manuscript carefully, I have some concerns which deny my recommendation for publication at the present form.

1. In the abstract, they mentioned that SURMOF technique is to improve the crystal quality, which is not true, at least in my opinion. SURMOF is actually for assembling high quality MOFs thin film on a substrate. This is highly required for fabrication of MOF-based devices. Indeed, if you only want to get MOF crystals with high quality, there are many useful methods, such as microwave synthesis, which have been demonstrated to work very well. In particular, for specific types of MOFs such as ZIFs, HKUST-1, Prussian Blue analogues, the reactions can be finished within minutes at room-temperature. In my opinion, there are no superiority for this technique in powder fabrication. It would be nice for the author to demonstrate the superiority of this technique in film fabrication, which unfortunately I didn't see clearly. In this work, the acoustomicrofluidic assembly indeed can fabricate films, but there are lack of characterizations of the films. For instance, the microstructure, the thickness, the crystallographic orientation at a film-state, the gas-sorption at a film-state, and the stability of the film are all necessarily to be provided. Unfortunately, the shape, the crystallographic orientation or the gas-sorption of the products all for the powders.

2. The activation step for MOFs are essential but not always, because it depends on the pore-size of the crystals and solvents used. For MOFs with large pores, such as MILs, the excessive ligands are usually included in the pores, therefore, long-term activation is necessary to generate empty pores. Or in some case, when reaction solvents are of high boiling point, such DMF, it is necessary to do solvent-exchange with solvent of low boiling point. The HKUST chosen in this manuscript is not an appropriate case, because its pore-size is small, therefore no ligands can be included. Furthermore, the solvent is mixed ethanol and water, which is easy to be removed. In most case of the ZIF crystals or others, we don't have to do activation. If the author can demonstrate a simultaneous-activation of MIL-88 or MIL-101 by this technique, it will be more convincing.

3. I noticed that the crystals fabricated have to be washed by EtOH/H₂O to remove the residuals, if so, it is not very significant for saving time.

4. I was surprised that the BET surface area of the crystals was evaluated by N₂ sorption at 298K according to their description “The BET and Langmuir surface areas were calculated from nitrogen physisorption measurements by placing approximately 0.5 g of the crystals in a surface area and porosity analyzer (ASAP 2020;Micromeritics Instrument Corp. Norcross, GA, USA) under N₂ at 25 oC for 12 hrs.”. I just don’t know how the N₂ molecules can be adsorbed in the pores of MOFs at such a high temperature. There is almost no adsorption between the surface of the MOF and N₂ at such a high temperature. The MOF society or porous solids society do N₂ sorption at 77 K according to any of the refs which we can get. Please have a check.

Reviewer #3 (Remarks to the Author):

In this work Yeo et al report the fabrication of highly oriented and simultaneously activated freestanding MOF crystals using acoustomicrofluidic technique. The fabricated MOF crystals were found to have the same properties similar to that of SURMOFs. These crystals were synthesized in a very fast way of about 5 minutes at room temperature. The authors managed to demonstrate that this technique can help to simultaneously synthesis and activate the MOF crystals in a single step. They also managed to have very good control over the out-of-plane architecture of the crystals through the manipulating the power intensity delivered to the device. I believe this work is providing an attractive platform for environmentally-friendly large-scale MOF production.

The work is very well done and the conclusions are fully supported by the experimental results. Also, the paper is very well written and is suitable for publication.

I believe that is work will be more better if they could demonstrate thsi method can be used to prepare other types of MOFs to show the novelty of the method.

Specific Responses to the Comments of Reviewer #1

The authors reported a fast way to synthesize HKUST-1 by an acoustically-driven method, as a potential candidate for growing HKUST-1 (and others similar MOFs) rapidly (synthesis times around 5 min) with a great control over crystal size, orientation, and defects. The method described is an alternative to liquid-phase epitaxy pathway, that consists in surface crystallization of materials “layer-by-layer” in a solution in contact to a controlling surface (SURMOFS). The advantages are faster synthesis times and in-situ activation of the solvent molecules in the as-synthesized materials.

The findings are of importance for the MOF community as well as related disciplines (catalysis, gas-adsorption, sensing) as this new approach offers a fast and defined synthesis of MOF materials.

The results and discussion are convincing and the conclusions are clear.

I believe that this paper could lead to a general trend towards this alternative synthesis pathway.

We are extremely grateful to the reviewer for their careful reading of our work and are pleased to learn of their very positive review.

Minor aspects:

The XDR reflections for the acoustically-synthesized materials are shifted (+0.4° approx.). An explanation for this shift should be given. The Miller indexes from the XRD reflections do not correspond to the marked values if you do not consider that shift.

We thank the reviewer for their insightful remark. It is true that there is a slight shift towards higher 2θ due to the compressive stresses arising from the acoustic irradiation. We have recently reported similar effects on the lattice structure pertaining to salt crystals (Ahmed et al., *Advanced Materials* 30, 1602040, 2018). We have now added a sentence to the revised manuscript on page 7 to clarify this.

The temperatures given in Fig S4 do not correspond to the commented values in text. The temperature difference is not the same. The authors should clarify which what temperature they were using.

We apologise for this oversight and have corrected it in the text (from 27.8 to 28.6 °C).

Specific Responses to the Comments of Reviewer #2

The manuscript comes from Prof. Yeo and colleagues reported acoustofluidic assembly of MOFs. At the first glance, I was fascinated by the idea that the free-standing MOF film can be highly-oriented and simultaneously-activated. We all know that the highly-oriented/monocrystalline thin film of MOFs is essential for fabrication of MOF-based devices and is really challenging except the layer-by-layer growth. Furthermore, activation of many MOFs is really time-consuming and make many drawbacks in improve the performance of MOF-based devices. If this manuscript can address the two problems by acoustofluidic assembly, it is definitely a big step towards practical application of MOFs.

However, after evaluation of this manuscript carefully, I have some concerns which deny my recommendation for publication at the present form.

We thank the reviewer for their thorough scrutiny of our work. It does appear from the second sentence above that the reviewer has misunderstood the original premise of our work, which was not to synthesize freestanding MOF films but rather that of highly-oriented powders. We apologise for this. Perhaps this was due to the parallels we drew with SURMOFs (our intention there was only to suggest that the highly-oriented crystals we obtain were akin to that seen in SURMOFs rather than to infer any direct comparison or to indicate we were synthesizing films) and our use of the term 'freestanding'. We have now removed any reference to the latter from the title and the manuscript to avoid confusion, and shall further elaborate on these points below, in addition to defending our claims to the superiority of our technique for *powder* MOF production, namely, the capability for simultaneous activation.

1. In the abstract, they mentioned that SURMOF technique is to improve the crystal quality, which is not true, at least in my opinion. SURMOF is actually for assembling high quality MOFs thin film on a substrate. This is highly required for fabrication of MOF-based devices. Indeed, if you only want to get MOF crystals with high quality, there are many useful methods, such as microwave synthesis, which have been demonstrated to work very well. In particular, for specific types of MOFs such as ZIFs, HKUST-1, Prussian Blue analogues, the reactions can be finished within minutes at room-temperature.

We concur with the reviewer that the LBL SURMOF technique's aim is to assemble high quality films rather than to improve the quality of the crystals themselves, for which, we agree with the reviewer, there are many other methods. When we mentioned that layer-by-layer deposition in the SURMOF technique offers structural control during crystal growth in the abstract, we were indeed referring to the film quality and their orientation, rather than the quality of the crystals themselves, e.g., absence of defects.

It is, however, important to mention here that we are NOT demonstrating our technique for the fabrication of MOF *films* like that of SURMOFs. Rather, the primary intent of the work was to show that the acoustic platform was capable of producing highly-oriented and simultaneously-activated *powdered* MOFs—a feat not yet accomplished by any other technique to our knowledge.

In my opinion, there are no superiority for this technique in powder fabrication. It would be nice for the author to demonstrate the superiority of this technique in film fabrication, which unfortunately I didn't see clearly. In this work, the acoustofluidic assembly indeed can fabricate films, but there are lack of characterizations of the films. For instance, the microstructure, the thickness, the crystallographic orientation at a film-state, the gas-sorption at a film-state, and the stability of the film are all necessarily to be provided. Unfortunately, the shape, the crystallographic orientation or the gas-sorption of the products all for the powders.

We apologise if we misled the reviewer. As mentioned above, our claim was not to demonstrate the formation of *films*. Rather, the main focus of our work is to show the production of highly-oriented *powdered* MOFs that are simultaneously activated. The characterization was therefore performed for powdered samples and hence not films. To avoid this confusion, we have added the word 'powder' and removed the term 'freestanding' from the manuscript.

To our knowledge, this is the first time highly-oriented powdered MOFs that do not require subsequent chemical or thermal activation steps is reported. This was grasped by Reviewers #1 and #3 who noted the importance and potential impact of the work. As such, we politely disagree with the reviewer that this work is not superior to other powder fabrication techniques since no other method shows the ability for simultaneous activation. In addition to the highly-oriented structures enhancing the availability and specificity of the MOFs towards guest species, such simultaneous activation of the MOFs without requiring chemical or thermal procedures (e.g., conventional heating and vacuum, solvent-exchange, supercritical CO₂ exchange, freeze-drying, chemical treatment), which are not only time consuming but also challenging, allows the MOFs to be truly classified as 'green materials' through a fast and environmentally-friendly alternative synthesis process.

2. The activation step for MOFs are essential but not always, because it depends on the pore-size of the crystals and solvents used. For MOFs with large pores, such as MILs, the excessive ligands are usually included in the pores, therefore, long-term activation is necessary to generate empty pores. Or in some case, when reaction solvents are of high boiling point, such DMF, it is necessary to do solvent-exchange with solvent of low boiling point. The HKUST chosen in this manuscript is not an appropriate case, because its pore-size is small, therefore no ligands can be included. Furthermore, the solvent is mixed ethanol and water, which is easy to be removed. In most case of the ZIF crystals or others, we don't have to do activation. If the author can demonstrate a simultaneous-activation of MIL-88 or MIL-101 by this technique, it will be more convincing.

We believe the reviewer may have misunderstood us when we claimed 'simultaneous activation'. MOF activation does not just refer to the removal of solvents from their pores, which the reviewer refers to, but also the removal of unreacted ligands and metals, which also requires subsequent washing steps (Farha & Hupp, *Accounts of Chemical Research* 43, 1166, 2010). As such, we politely disagree with the reviewer that HKUST-1 is not an appropriate model. There are many instances in the literature which report on the importance of 'free-ing' HKUST-1 pores from residual solvents and ligands in the reaction

mixture (see, for example, Kim et al., *Journal of the American Chemical Society* 137, 10009, 2015; Bordiga et al., *Physical Chemistry Chemical Physics* 9, 2676, 2007; Prestipino et al., *Chemistry of Materials* 18, 1337, 2006; among others).

That said, we have nevertheless followed the recommendation of the reviewer to provide another convincing demonstration of simultaneous activation using MIL-88B as a model. As with HKUST-1, we observe a highly-oriented structure at the highest input voltage, as shown in Fig. 5 in the revised manuscript.

On the reviewer's comments regarding the low boiling point of the solvent used in our HKUST-1 MOF synthesis, similar surface areas were obtained when the synthesis was carried out with a high boiling point solvent (DMF), as shown in the figure below.

3. I noticed that the crystals fabricated have to be washed by EtOH/H₂O to remove the residuals, if so, it is not very significant for saving time.

We agree with the reviewer that the washing steps are not of particular importance for the synthesis and characterization of the MOFs produced in our work. However, we added this step to remove excess reactants remaining in the subsequent powder in order to accurately analyse their mass. Moreover, this is a common protocol for many applications, particularly, sensing. Nevertheless, we have removed the washing step and observed no significant change in the surface area.

4. I was surprised that the BET surface area of the crystals was evaluated by N₂ sorption at 298K according to their description “The BET and Langmuir surface areas were calculated from nitrogen physisorption measurements by placing approximately 0.5 g of the crystals in a surface area and porosity analyzer (ASAP 2020;Micromeritics Instrument Corp. Norcross, GA, USA) under N₂ at 25 oC for 12 hrs.”. I just don’t know how the N₂ molecules can be adsorbed in the pores of MOFs at such a high temperature. There is almost no adsorption between the surface of the MOF and N₂ at such a high temperature. The MOF society or porous solids society do N₂ sorption at 77 K according to any of the refs which we can get. Please have a check.

We thank the reviewer for picking up on this and apologise for the mistake in explaining the procedure. What should have been stated is that “The sample was left for a degassing step at 25°C for 12 hrs prior to running the N₂ physisorption measurement at 77K”. This has now been corrected in the experimental methods section of the revised manuscript. We are also including with this rebuttal one of the BET reports stating that adsorption/desorption of N₂ was indeed carried out at 77K.

Specific Responses to the Comments of Reviewer #3

In this work Yeo et al report the fabrication of highly oriented and simultaneously activated freestanding MOF crystals using acoustofluidic technique. The fabricated MOF crystals were found to have the same properties similar to that of SURMOFs. These crystals were synthesized in a very fast way of about 5 minutes at room temperature. The authors managed to demonstrate that this technique can help to simultaneously synthesis and activate the MOF crystals in a single step. They also managed to have very good control over the out-of-plane architecture of the crystals through the manipulating the power intensity delivered to the device. I believe this work is providing an attractive platform for environmentally-friendly large-scale MOF production.

The work is very well done and the conclusions are fully supported by the experimental results. Also, the paper is very well written and is suitable for publication.

We are extremely grateful to the reviewer for their careful reading of our work and are pleased to learn of their very positive review.

I believe that is work will be more better if they could demonstrate thsi method can be used to prepare other types of MOFs to show the novelty of the method.

As suggested by the reviewer, we have now included the results for the synthesis of another MOF (MIL-88B).

Sample: 30mv after
Operator: HA
Submitter: Heba Ahmed
File: E:\SI0127.SMP

Started: 1/09/2017 12:55:46 PM	Analysis Adsorptive: N2
Completed: 2/09/2017 1:22:24 PM	Analysis Bath Temp.: 77.150 K
Report Time: 4/09/2017 11:54:09 AM	Thermal Correction: No
Sample Mass: 1.4443 g	Cold Free Space: 92.5921 cm ³ Measured
Equilibration Interval: 15 s	Low Pressure Dose: None
Sample Density: 1.000 g/cm ³	Automatic Degas: No

Summary Report

Surface Area

Single point surface area at $p/p^\circ = 0.303270013$: 1,390.0762 m²/g

BET Surface Area: 1,598.4200 m²/g

Langmuir Surface Area: 2,054.6573 m²/g

t-Plot Micropore Area: 1,364.0147 m²/g

t-Plot External Surface Area: 234.4052 m²/g

BJH Adsorption cumulative surface area of pores
between 17.000 Å and 3,000.000 Å width: 189.397 m²/g

BJH Desorption cumulative surface area of pores
between 17.000 Å and 3,000.000 Å width: 125.0515 m²/g

Pore Volume

Single point adsorption total pore volume of pores
less than 11.415 Å width at $p/p^\circ = 0.010000000$: 0.643418 cm³/g

t-Plot micropore volume: 0.596145 cm³/g

BJH Adsorption cumulative volume of pores
between 17.000 Å and 3,000.000 Å width: 0.119832 cm³/g

BJH Desorption cumulative volume of pores
between 17.000 Å and 3,000.000 Å width: 0.089394 cm³/g

Pore Size

Adsorption average pore diameter (4V/A by BET): 16.1013 Å

BJH Adsorption average pore width (4V/A): 25.308 Å

BJH Desorption average pore width (4V/A): 28.594 Å

Sample: 30mv after
 Operator: HA
 Submitter: Heba Ahmed
 File: E:\SI0127.SMP

Started: 1/09/2017 12:55:46 PM	Analysis Adsorptive: N2
Completed: 2/09/2017 1:22:24 PM	Analysis Bath Temp.: 77.150 K
Report Time: 4/09/2017 11:54:09 AM	Thermal Correction: No
Sample Mass: 1.4443 g	Cold Free Space: 92.5921 cm ³ Measured
Equilibration Interval: 15 s	Low Pressure Dose: None
Sample Density: 1.000 g/cm ³	Automatic Degas: No

Isotherm Tabular Report

Relative Pressure (p/p°)	Absolute Pressure (kPa)	Quantity Adsorbed (mmol/g)	Elapsed Time (h:min)	Saturation Pressure (kPa)
			11:36	102.0149821
0.009549352	0.9737656	18.55118	12:30	
0.029934243	3.0517115	18.86552	13:01	
0.058859375	5.9990915	19.29159	13:32	
			13:37	101.9184568
0.077811913	7.9292341	19.48003	13:53	
0.100442282	10.2332327	19.64854	14:14	
0.123547968	12.5849492	19.79473	14:33	
0.145364342	14.8041967	19.92197	14:54	
0.166933604	16.9975390	20.03129	15:14	
0.188763916	19.2167875	20.12457	15:33	
			15:42	101.7943540
0.210505012	21.4273220	20.20333	15:57	
0.248916713	25.3359045	20.31987	16:16	
0.303270013	30.8665951	20.44769	16:35	
0.362197051	36.8623904	20.55644	16:52	
0.394025027	40.1003166	20.60615	17:04	
0.444888678	45.2749859	20.67584	17:18	
0.495832801	50.4574385	20.73590	17:32	
			17:43	101.7598760
0.545397445	55.5000425	20.78798	17:44	
0.595713827	60.6258754	20.83438	17:55	
0.644758827	65.6227028	20.87766	18:05	
0.695817586	70.8259382	20.91957	18:16	
0.736404056	74.9628223	20.94924	18:25	
0.766868529	78.0685653	20.96924	18:32	
0.797160451	81.1571060	20.98634	18:39	
0.817153625	83.1960582	20.99780	18:44	
0.837898544	85.3124374	21.00850	18:50	
0.857634662	87.3255780	21.01801	18:55	
0.872898520	88.8827501	21.02514	18:59	
0.887815950	90.4055093	21.03250	19:04	
0.902993177	91.9540802	21.03967	19:08	
0.913100517	92.9864581	21.04461	19:12	
0.923214927	94.0188360	21.04941	19:15	
0.931301462	94.8447449	21.05350	19:18	
0.938373813	95.5674070	21.05755	19:21	
0.945446846	96.2901748	21.06159	19:24	
0.951422306	96.9011924	21.06474	19:27	
0.957144039	97.4863984	21.06867	19:30	
0.962696432	98.0543856	21.07199	19:33	
0.966228876	98.4158306	21.07492	19:35	
0.969837666	98.7858930	21.07794	19:38	
0.972516869	99.0612855	21.08032	19:41	
0.975533850	99.3710989	21.08361	19:44	101.8633018
0.979701522	99.7927847	21.08700	19:47	
0.981334855	99.9563049	21.08937	19:50	

Sample: 30mv after
 Operator: HA
 Submitter: Heba Ahmed
 File: E:\SI0127.SMP

Started: 1/09/2017 12:55:46 PM
 Completed: 2/09/2017 1:22:24 PM
 Report Time: 4/09/2017 11:54:09 AM
 Sample Mass: 1.4443 g
 Equilibration Interval: 15 s
 Sample Density: 1.000 g/cm³

Analysis Adsorptive: N2
 Analysis Bath Temp.: 77.150 K
 Thermal Correction: No
 Cold Free Space: 92.5921 cm³ Measured
 Low Pressure Dose: None
 Automatic Degas: No

Isotherm Tabular Report

Relative Pressure (p/p ^o)	Absolute Pressure (kPa)	Quantity Adsorbed (mmol/g)	Elapsed Time (h:min)	Saturation Pressure (kPa)
0.983982029	100.2230799	21.09258	19:53	
0.985446601	100.3693895	21.09527	19:56	
0.987915666	100.6189540	21.09889	19:58	
0.988619871	100.6878041	21.10070	20:01	
0.990253676	100.8513242	21.10370	20:04	
0.993915580	101.2213785	21.11073	20:07	
0.976037932	99.3969188	21.09468	20:11	
0.959013080	97.6585115	21.08280	20:16	
0.938278374	95.5415953	21.07093	20:22	
0.934418588	95.1458514	21.06781	20:25	
0.924897477	94.1736899	21.06362	20:28	
0.916896794	93.3563904	21.05943	20:31	
0.905946802	92.2379762	21.05433	20:35	
0.891708917	90.7840427	21.04783	20:40	
0.876616301	89.2440811	21.04138	20:44	
0.861606946	87.7127126	21.03543	20:48	
0.842471440	85.7597965	21.02663	20:54	
0.821889116	83.6606279	21.01692	20:59	
0.802073150	81.6388861	21.00756	21:05	
0.773217723	78.6965981	20.99305	21:12	
0.743013045	75.6166668	20.97741	21:20	
0.704112213	71.6515785	20.95517	21:29	
0.654986724	66.6461459	20.92344	21:39	
			21:45	101.7460914
0.604606348	61.5117199	20.88716	21:50	
0.554568978	56.4116865	20.84517	22:01	
0.504418807	51.3010908	20.79659	22:13	
0.445039983	45.2491620	20.68719	22:32	
0.400920610	40.7547877	20.62452	22:46	
0.341553466	34.7095152	20.52899	23:06	
0.303810810	30.8665951	20.45735	23:22	
0.255132810	25.9139963	20.34747	23:40	
			23:47	101.5599331
0.203415523	20.6588670	20.19685	24:02	
0.146363271	14.8646440	19.95102	24:30	

Sample: 30mv after
 Operator: HA
 Submitter: Heba Ahmed
 File: E:\SI0127.SMP

Started: 1/09/2017 12:55:46 PM
 Completed: 2/09/2017 1:22:24 PM
 Report Time: 4/09/2017 11:54:09 AM
 Sample Mass: 1.4443 g
 Equilibration Interval: 15 s
 Sample Density: 1.000 g/cm³

Analysis Adsorptive: N2
 Analysis Bath Temp.: 77.150 K
 Thermal Correction: No
 Cold Free Space: 92.5921 cm³ Measured
 Low Pressure Dose: None
 Automatic Degas: No

Isotherm Linear Plot

Sample: 30mv after
 Operator: HA
 Submitter: Heba Ahmed
 File: E:\SI0127.SMP

Started: 1/09/2017 12:55:46 PM	Analysis Adsorptive: N2
Completed: 2/09/2017 1:22:24 PM	Analysis Bath Temp.: 77.150 K
Report Time: 4/09/2017 11:54:09 AM	Thermal Correction: No
Sample Mass: 1.4443 g	Cold Free Space: 92.5921 cm ³ Measured
Equilibration Interval: 15 s	Low Pressure Dose: None
Sample Density: 1.000 g/cm ³	Automatic Degas: No

BET Surface Area Report

BET Surface Area: 1598.4200 ± 27.8433 m²/g
 Slope: 0.061337 ± 0.001056 g/mmol
 Y-Intercept: -0.000293 ± 0.000122 g/mmol
 C: -208.041249
 Qm: 16.38178 mmol/g
 Correlation Coefficient: 0.9989636
 Molecular Cross-Sectional Area: 0.1620 nm²

Relative Pressure (p/p ^o)	Quantity Adsorbed (mmol/g)	1/[Q(p ^o /p - 1)]
0.009549352	18.55118	0.00052
0.029934243	18.86552	0.00164
0.058859375	19.29159	0.00324
0.077811913	19.48003	0.00433
0.100442282	19.64854	0.00568
0.123547968	19.79473	0.00712
0.145364342	19.92197	0.00854
0.166933604	20.03129	0.01000
0.188763916	20.12457	0.01156

Sample: 30mv after
 Operator: HA
 Submitter: Heba Ahmed
 File: E:\SI0127.SMP

Started: 1/09/2017 12:55:46 PM
 Completed: 2/09/2017 1:22:24 PM
 Report Time: 4/09/2017 11:54:09 AM
 Sample Mass: 1.4443 g
 Equilibration Interval: 15 s
 Sample Density: 1.000 g/cm³

Analysis Adsorptive: N2
 Analysis Bath Temp.: 77.150 K
 Thermal Correction: No
 Cold Free Space: 92.5921 cm³ Measured
 Low Pressure Dose: None
 Automatic Degas: No

BET Surface Area Plot

Sample: 30mv after
 Operator: HA
 Submitter: Heba Ahmed
 File: E:\SI0127.SMP

Started: 1/09/2017 12:55:46 PM	Analysis Adsorptive: N2
Completed: 2/09/2017 1:22:24 PM	Analysis Bath Temp.: 77.150 K
Report Time: 4/09/2017 11:54:09 AM	Thermal Correction: No
Sample Mass: 1.4443 g	Cold Free Space: 92.5921 cm ³ Measured
Equilibration Interval: 15 s	Low Pressure Dose: None
Sample Density: 1.000 g/cm ³	Automatic Degas: No

t-Plot Report

Micropore Volume: 0.596145 cm³/g
 Micropore Area: 1364.0147 m²/g
 External Surface Area: 234.4052 m²/g
 Slope: 0.676101 ± 0.036969 mmol/g·Å
 Y-Intercept: 17.194785 ± 0.154188 mmol/g
 Correlation Coefficient: 0.992608
 Surface Area Correction Factor: 1.000
 Density Conversion Factor: 0.0015468
 Total Surface Area (BET): 1598.4200 m²/g
 Thickness Range: 3.5000 Å to 5.0000 Å
 Thickness Equation: Harkins and Jura

Thickness Curve

$$t = [13.99 / (0.034 - \log(p/p^\circ))] ^{0.5}$$

t-Plot Report - Data

Relative Pressure (p/p [°])	Statistical Thickness (Å)	Quantity Adsorbed (mmol/g)	Fitted
0.029934243	2.9967	18.86552	
0.058859375	3.3266	19.29159	
0.077811913	3.4986	19.48003	
0.100442282	3.6817	19.64854	*
0.123547968	3.8534	19.79473	*
0.145364342	4.0065	19.92197	*
0.166933604	4.1522	20.03129	*
0.188763916	4.2959	20.12457	*
0.210505012	4.4366	20.20333	*
0.248916713	4.6829	20.31987	*
0.303270013	5.0335	20.44769	
0.362197051	5.4267	20.55644	
0.394025027	5.6485	20.60615	
0.444888678	6.0222	20.67584	
0.495832801	6.4272	20.73590	
0.545397445	6.8600	20.78798	
0.595713827	7.3501	20.83438	
0.644758827	7.8923	20.87766	
0.695817586	8.5471	20.91957	
0.736404056	9.1559	20.94924	
0.766868529	9.6808	20.96924	
0.797160451	10.2772	20.98634	
0.817153625	10.7219	20.99780	
0.837898544	11.2363	21.00850	
0.857634662	11.7869	21.01801	
0.872898520	12.2626	21.02514	
0.887815950	12.7784	21.03250	
0.902993177	13.3655	21.03967	
0.913100517	13.7981	21.04461	
0.923214927	14.2705	21.04941	
0.931301462	14.6809	21.05350	

Sample: 30mv after
 Operator: HA
 Submitter: Heba Ahmed
 File: E:\SI0127.SMP

Started: 1/09/2017 12:55:46 PM
 Completed: 2/09/2017 1:22:24 PM
 Report Time: 4/09/2017 11:54:09 AM
 Sample Mass: 1.4443 g
 Equilibration Interval: 15 s
 Sample Density: 1.000 g/cm³

Analysis Adsorptive: N2
 Analysis Bath Temp.: 77.150 K
 Thermal Correction: No
 Cold Free Space: 92.5921 cm³ Measured
 Low Pressure Dose: None
 Automatic Degas: No

t-Plot Report - Data

Relative Pressure (p/p°)	Statistical Thickness (Å)	Quantity Adsorbed (mmol/g)	Fitted
0.938373813	15.0672	21.05755	
0.945446846	15.4825	21.06159	
0.951422306	15.8587	21.06474	
0.957144039	16.2434	21.06867	
0.962696432	16.6425	21.07199	
0.966228876	16.9109	21.07492	
0.969837666	17.1978	21.07794	
0.972516869	17.4199	21.08032	
0.975533850	17.6797	21.08361	
0.979701522	18.0571	21.08700	
0.981334855	18.2113	21.08937	
0.983982029	18.4692	21.09258	
0.985446601	18.6164	21.09527	
0.987915666	18.8722	21.09889	
0.988619871	18.9470	21.10070	
0.990253676	19.1237	21.10370	
0.993915580	19.5375	21.11073	

Sample: 30mv after
 Operator: HA
 Submitter: Heba Ahmed
 File: E:\SI0127.SMP

Started: 1/09/2017 12:55:46 PM
 Completed: 2/09/2017 1:22:24 PM
 Report Time: 4/09/2017 11:54:09 AM
 Sample Mass: 1.4443 g
 Equilibration Interval: 15 s
 Sample Density: 1.000 g/cm³

Analysis Adsorptive: N2
 Analysis Bath Temp.: 77.150 K
 Thermal Correction: No
 Cold Free Space: 92.5921 cm³ Measured
 Low Pressure Dose: None
 Automatic Degas: No

Sample: 30mv after
 Operator: HA
 Submitter: Heba Ahmed
 File: E:\SI0127.SMP

Started: 1/09/2017 12:55:46 PM
 Completed: 2/09/2017 1:22:24 PM
 Report Time: 4/09/2017 11:54:09 AM
 Sample Mass: 1.4443 g
 Equilibration Interval: 15 s
 Sample Density: 1.000 g/cm³

Analysis Adsorptive: N2
 Analysis Bath Temp.: 77.150 K
 Thermal Correction: No
 Cold Free Space: 92.5921 cm³ Measured
 Low Pressure Dose: None
 Automatic Degas: No

BJH Adsorption Pore Distribution Report

Faas Correction

Halsey

$$t = 3.54 [-5 / \ln(p/p^0)] ^{0.333}$$

Width Range: 17.000 Å to 3,000.000 Å

Adsorbate Property Factor: 9.53000 Å

Density Conversion Factor: 0.0015468

Fraction of Pores Open at Both Ends: 0.00

Pore Width Range (Å)	Average Width (Å)	Incremental Pore Volume (cm ³ /g)	Cumulative Pore Volume (cm ³ /g)	Incremental Pore Area (m ² /g)	Cumulative Pore Area (m ² /g)
3189.2 - 2002.5	2334.7	0.000256	0.000256	0.004	0.004
2002.5 - 1718.9	1838.8	0.000110	0.000366	0.002	0.007
1718.9 - 1620.2	1666.6	0.000066	0.000432	0.002	0.008
1620.2 - 1349.5	1459.9	0.000133	0.000565	0.004	0.012
1349.5 - 1228.2	1283.0	0.000100	0.000665	0.003	0.015
1228.2 - 1057.0	1129.5	0.000119	0.000784	0.004	0.019
1057.0 - 973.6	1011.8	0.000088	0.000872	0.003	0.023
973.6 - 810.9	877.1	0.000126	0.000999	0.006	0.029
810.9 - 723.8	762.3	0.000124	0.001123	0.007	0.035
723.8 - 661.0	689.4	0.000090	0.001213	0.005	0.040
661.0 - 592.0	622.5	0.000115	0.001329	0.007	0.048
592.0 - 537.3	561.9	0.000113	0.001441	0.008	0.056
537.3 - 469.4	498.6	0.000128	0.001569	0.010	0.066
469.4 - 415.6	439.1	0.000154	0.001723	0.014	0.080
415.6 - 371.3	390.9	0.000124	0.001846	0.013	0.093
371.3 - 329.9	348.1	0.000161	0.002007	0.019	0.111
329.9 - 297.0	311.6	0.000164	0.002171	0.021	0.132
297.0 - 266.6	280.1	0.000167	0.002338	0.024	0.156
266.6 - 236.5	249.7	0.000198	0.002536	0.032	0.188
236.5 - 212.7	223.2	0.000207	0.002744	0.037	0.225
212.7 - 184.8	196.6	0.000307	0.003051	0.062	0.287
184.8 - 163.7	172.9	0.000323	0.003374	0.075	0.362
163.7 - 146.7	154.2	0.000318	0.003692	0.082	0.445
146.7 - 129.3	136.8	0.000436	0.004127	0.127	0.572
129.3 - 115.0	121.2	0.000504	0.004631	0.166	0.738
115.0 - 103.9	108.8	0.000556	0.005187	0.205	0.943
103.9 - 90.6	96.2	0.000851	0.006039	0.354	1.297
90.6 - 80.2	84.7	0.001038	0.007077	0.490	1.787
80.2 - 69.5	74.0	0.001606	0.008683	0.868	2.655
69.5 - 59.3	63.5	0.002383	0.011065	1.501	4.156
59.3 - 51.9	55.0	0.002573	0.013639	1.871	6.028
51.9 - 45.7	48.3	0.002874	0.016512	2.378	8.406
45.7 - 40.8	42.9	0.003399	0.019911	3.169	11.574
40.8 - 36.5	38.4	0.004117	0.024029	4.294	15.869
36.5 - 32.9	34.4	0.005045	0.029074	5.860	21.729
32.9 - 30.8	31.7	0.003753	0.032827	4.729	26.458
30.8 - 27.4	28.8	0.008693	0.041519	12.061	38.518
27.4 - 24.5	25.8	0.010953	0.052472	17.009	55.527
24.5 - 22.7	23.5	0.010643	0.063115	18.111	73.639

Sample: 30mv after
 Operator: HA
 Submitter: Heba Ahmed
 File: E:\SI0127.SMP

Started: 1/09/2017 12:55:46 PM	Analysis Adsorptive: N2
Completed: 2/09/2017 1:22:24 PM	Analysis Bath Temp.: 77.150 K
Report Time: 4/09/2017 11:54:09 AM	Thermal Correction: No
Sample Mass: 1.4443 g	Cold Free Space: 92.5921 cm ³ Measured
Equilibration Interval: 15 s	Low Pressure Dose: None
Sample Density: 1.000 g/cm ³	Automatic Degas: No

Pore Width Range (Å)	Average Width (Å)	Incremental Pore Volume (cm ³ /g)	Cumulative Pore Volume (cm ³ /g)	Incremental Pore Area (m ² /g)	Cumulative Pore Area (m ² /g)
22.7 - 21.6	22.1	0.007492	0.070607	13.547	87.186
21.6 - 20.6	21.1	0.009226	0.079833	17.497	104.683
20.6 - 19.6	20.1	0.011225	0.091058	22.367	127.050
19.6 - 18.6	19.1	0.013404	0.104462	28.139	155.189
18.6 - 17.5	18.0	0.015370	0.119832	34.208	189.397

Sample: 30mv after
 Operator: HA
 Submitter: Heba Ahmed
 File: E:\SI0127.SMP

Started: 1/09/2017 12:55:46 PM
 Completed: 2/09/2017 1:22:24 PM
 Report Time: 4/09/2017 11:54:09 AM
 Sample Mass: 1.4443 g
 Equilibration Interval: 15 s
 Sample Density: 1.000 g/cm³

Analysis Adsorptive: N2
 Analysis Bath Temp.: 77.150 K
 Thermal Correction: No
 Cold Free Space: 92.5921 cm³ Measured
 Low Pressure Dose: None
 Automatic Degas: No

BJH Adsorption dV/dlog(w) Pore Volume

Halsey : Faas Correction

Sample: 30mv after
 Operator: HA
 Submitter: Heba Ahmed
 File: E:\SI0127.SMP

Started: 1/09/2017 12:55:46 PM	Analysis Adsorptive: N2
Completed: 2/09/2017 1:22:24 PM	Analysis Bath Temp.: 77.150 K
Report Time: 4/09/2017 11:54:09 AM	Thermal Correction: No
Sample Mass: 1.4443 g	Cold Free Space: 92.5921 cm ³ Measured
Equilibration Interval: 15 s	Low Pressure Dose: None
Sample Density: 1.000 g/cm ³	Automatic Degas: No

BJH Desorption Pore Distribution Report

Faas Correction

Halsey

$$t = 3.54 [-5 / \ln(p/p^0)] ^{0.333}$$

Width Range: 17.000 Å to 3,000.000 Å

Adsorbate Property Factor: 9.53000 Å

Density Conversion Factor: 0.0015468

Fraction of Pores Open at Both Ends: 0.00

Pore Width Range (Å)	Average Width (Å)	Incremental Pore Volume (cm ³ /g)	Cumulative Pore Volume (cm ³ /g)	Incremental Pore Area (m ² /g)	Cumulative Pore Area (m ² /g)
3189.2 - 827.6	969.1	0.000608	0.000608	0.025	0.025
827.6 - 490.2	575.3	0.000457	0.001065	0.032	0.057
490.2 - 329.4	378.1	0.000472	0.001537	0.050	0.107
329.4 - 310.6	319.5	0.000127	0.001664	0.016	0.123
310.6 - 272.4	288.9	0.000169	0.001833	0.023	0.146
272.4 - 247.0	258.4	0.000173	0.002006	0.027	0.173
247.0 - 219.1	231.3	0.000213	0.002219	0.037	0.210
219.1 - 191.2	203.1	0.000277	0.002496	0.054	0.264
191.2 - 168.5	178.3	0.000280	0.002776	0.063	0.327
168.5 - 150.8	158.6	0.000262	0.003037	0.066	0.393
150.8 - 133.0	140.6	0.000401	0.003439	0.114	0.507
133.0 - 118.0	124.5	0.000454	0.003893	0.146	0.653
118.0 - 106.4	111.5	0.000448	0.004341	0.161	0.814
106.4 - 93.1	98.8	0.000716	0.005057	0.290	1.104
93.1 - 82.3	86.9	0.000798	0.005855	0.367	1.471
82.3 - 71.5	76.0	0.001183	0.007038	0.623	2.094
71.5 - 61.2	65.4	0.001777	0.008815	1.087	3.181
61.2 - 53.1	56.4	0.002137	0.010952	1.515	4.695
53.1 - 46.8	49.4	0.002611	0.013563	2.112	6.808
46.8 - 41.6	43.8	0.003181	0.016744	2.905	9.713
41.6 - 36.5	38.6	0.007974	0.024718	8.254	17.967
36.5 - 33.3	34.7	0.004471	0.029189	5.148	23.116
33.3 - 29.6	31.1	0.007189	0.036378	9.233	32.348
29.6 - 27.4	28.4	0.005681	0.042059	8.007	40.356
27.4 - 24.9	26.0	0.009214	0.051273	14.192	54.548
24.9 - 22.3	23.4	0.013585	0.064858	23.212	77.760
22.3 - 19.7	20.8	0.024536	0.089394	47.291	125.052

Sample: 30mv after
 Operator: HA
 Submitter: Heba Ahmed
 File: E:\SI0127.SMP

Started: 1/09/2017 12:55:46 PM
 Completed: 2/09/2017 1:22:24 PM
 Report Time: 4/09/2017 11:54:09 AM
 Sample Mass: 1.4443 g
 Equilibration Interval: 15 s
 Sample Density: 1.000 g/cm³

Analysis Adsorptive: N2
 Analysis Bath Temp.: 77.150 K
 Thermal Correction: No
 Cold Free Space: 92.5921 cm³ Measured
 Low Pressure Dose: None
 Automatic Degas: No

BJH Desorption dV/dlog(w) Pore Volume

Halsey : Faas Correction

Reviewers' comments:

Reviewer #1 (Remarks to the Author):

The authors provided a thorough revision of their manuscript. All critical points raised by the reviewers were

satisfactory addressed. The respective changes were included in the new version of the manuscript. From my side there are no concerns.

Reviewer #2 (Remarks to the Author):

Prof. Yeo revised this manuscript, and explained that this manuscript was not about MOF film, which certainly solved one of my biggest concern. However, another concern has been raised by this explanation. In the abstract part, the authors claimed that "Controlled epitaxial growth of surface anchored MOFs (SURMOFs), which involves successive deposition/growth of individual monolayers, has been proposed to improve crystal quality during synthesis, but is nevertheless a lengthy and tedious process." But according to my knowledge, SURMOF is not for improving the quality of MOF crystals. SURMOF is to fabricate MOF THIN FILM not crystalline particles. I have copied the page link from the inventor of SURMOF, Prof. Christof Wöll. Please have a check (<https://www.ifg.kit.edu/english/2872.php>). In MOFs filed, we never use SURMOF to improve the quality of MOFs crystals. There is no point to compare this method with SURMOF unless MOFs film is the target.

I still can not see advantage of the reported method on improving the crystal quality. With conventional methods, we can always get high quality single crystals. I would like to emphasise that we already have the exact crystal structures of many MOFs. Most of the data was obtained by measure single crystal. The single crystals contain few defects otherwise we can not solve the structure by X-ray diffraction. Furthermore, I noticed that the particles have significant defects such as holes according to the SEM image. At least, this kind of particle can not meet the requirements for single crystal diffraction. I would suggest the authors to carefully compare their sample with single crystals.

Regarding the reaction speed, I admit that this method is much faster than solvothermal method. But there is no advantage when you compare the reaction speed to microwave synthesis. Microwave synthesis can fabricate MOFs crystals within several minutes.

The authors claimed they can control the orientation of the obtained MOFs. First, I would like to suggest the authors to get simulated diffraction patterns from cif files of single crystals. Second, it would be better to overlay the experimental data with the simulated data to see the differences. They obtained some samples, but with different diffraction peaks. It is not convincing to simply say that the different diffraction peaks come from different orientations. During the synthesis of MOFs, you may obtain byproducts or other phases, it would be nice if the authors can provide convincing evidence to demonstrate that the changed diffraction pattern really correlates to orientation. I would suggest the authors to perform DFT calculation to simulate the crystal structure, which is a general method when we want to know the structure change of the MOFs without single-crystal data. Third, I would like to suggest the authors to characterise gas sorption for all the samples with different diffraction profiles.

The author reported that they synthesized MIL-88. But, the diffraction pattern is not the same as the typical MIL-88, indicating either the MIL-88 is not activated, or the sample is not MIL-88.

We know that the samples fabricated in this method need to be washed to remove the byproducts, salts. Because the wash step is always, you can never really escape from re-activation steps.

I could not understand the term "highly oriented" in the title. Every MOF single crystal is highly oriented, because every single crystal is oriented by definition of crystals.

Reviewer #3 (Remarks to the Author):

I am satisfied with the changes made in the revised version of the paper and I recommend it for publication with no further changes.

Specific Responses to Reviewer #2

Prof. Yeo revised this manuscript, and explained that this manuscript was not about MOF film, which certainly solved one of my biggest concern. However, another concern has been raised by this explanation. In the abstract part, the authors claimed that "Controlled epitaxial growth of surface anchored MOFs (SURMOFs), which involves successive deposition/growth of individual monolayers, has been proposed to improve crystal quality during synthesis, but is nevertheless a lengthy and tedious process." But according to my knowledge, SURMOF is not for improving the quality of MOF crystals. SURMOF is to fabricate MOF THIN FILM not crystalline particles. I have copied the page link from the inventor of SURMOF, Prof. Christof Wöll. Please have a check (<https://www.ifg.kit.edu/english/2872.php>). In MOFs filed, we never use SURMOF to improve the quality of MOFs crystals. There is no point to compare this method with SURMOF unless MOFs film is the target.

We thank the reviewer for their re-evaluation of our work and are pleased to learn that we have solved the reviewer's biggest concern. We are pleased that it is now clear that the premise of our work is not to produce MOF thin films, but rather to synthesize MOF powders.

We agree with the reviewer that SURMOFs are generally used to produce thin films of high orientation, and not for improving the crystal quality. We wish to clarify that our previous comparison with epitaxial growth methods was intended only to suggest that the ability of our synthesis method to produce MOFs with orientation *similar* to many epitaxially grown SURMOFs reported in the literature (see, for example, Shekhah, *Chem Commun* 2013, 49, 10079; Bux, *Chem Mater* 2011, 23, 2262 and Li, *Nano Res*, 2018, 11, 1850, to name a few). However, we accept that such a comparison may confuse the reader and hence have now removed the comparison with SURMOFs throughout the revised manuscript.

I still can not see advantage of the reported method on improving the crystal quality. With conventional methods, we can always get high quality single crystals. I would like to emphasise that we already have the exact crystal structures of many MOFs. Most of the data was obtained by measure single crystal. The single crystals contain few defects otherwise we can not solve the structure by X-ray diffraction. Furthermore, I noticed that the particles have significant defects such as holes according to the SEM image. At least, this kind of particle can not meet the requirements for single crystal diffraction. I would suggest the authors to carefully compare their sample with single crystals.

Given that our work reports the synthesis of *powder* MOFs, we feel that it is tenuous to compare the quality of our powder samples with that of SURMOF films.

We agree with the reviewer that our crystals might contain some defects, but this could be due to post-synthesis handling. We note that it is not unusual for MOF crystals to exhibit post-synthesis defects such as dislocations and fractures, especially along the {111} plane, due to the fragility of these materials. This has been explained by Maryiam et al., *CrystEngComm*, 2008, 10, 646 who reported that cleavage occurring early on after crystallization is complete is probably caused by the crystal fracturing during post-synthesis treatment. It is therefore

likely that the formation of fractures along the {110} plane occurs in the {111} facets of the crystal because the crystal planes within the structure are possible fracture planes that intersect with these {111} facets.

Whatever the reason for the defects, we nevertheless wish to stress that the aim of our work is to show the ability to synthesize MOF powders (not films) that are oriented and simultaneously activated (not of better *quality*, in terms of defects). As such, we do not make any claims regarding crystal *quality* anywhere in the manuscript. Additionally, we have compared our powder XRD data for HKUST-1 at different powers to DFT-modeled CIF files showing slight strain occurring on the crystal during synthesis.

Regarding the reaction speed, I admit that this method is much faster than solvothermal method. But there is no advantage when you compare the reaction speed to microwave synthesis. Microwave synthesis can fabricate MOFs crystals within several minutes.

We agree with the reviewer on this point. While speed could be inferred given that the simplicity of the process could mean it can be carried out more quickly, we however do *not* make any overt claims regarding the speed of the process. Rather, the emphasis of our work centers around the fact that the powders we synthesize are highly oriented and simultaneously activated. To clarify this point, we have now de-emphasized any claims of the speed of synthesis and added on page 12 of the revised manuscript that the duration of synthesis is compared to microwave synthesis methods.

The authors claimed they can control the orientation of the obtained MOFs. First, I would like to suggest the authors to get simulated diffraction patterns from cif files of single crystals. Second, it would be better to overlay the experimental data with the simulated data to see the differences. They obtained some samples, but with different diffraction peaks. It is not convincing to simply say that the different diffraction peaks come from different orientations. During the synthesis of MOFs, you may obtain byproducts or other phases, it would nice if the authors can provide convincing evidence to demonstrate that the changed diffraction pattern really correlates to orientation. I would suggest the authors to perform DFT calculation to simulate the crystal structure, which is a general method when we want to know the structure change of the MOFs without single-crystal data. Third, I would like to suggest the authors to characterise gas sorption for all the samples with different diffraction profiles.

We thank the reviewer for this suggestion. We have now modelled our HKUST powder XRD pattern as a function of the input power against DFT-simulated CIF files, which show that these patterns are due to some strain effect on the crystals. In addition, we also correlated the MIL-88B data to the single crystal data obtained from the single crystal CIF file obtained from the CCDC database. For clarity, we have superimposed the fitted experimental vs simulated data.

Moreover, as per the reviewer's suggestion, we have also included the BET for other powers showing the increase in surface area with increasing input power.

They author reported that they synthesized MIL-88. But, the diffraction pattern is not the same as the typical MIL-88, indicating either the MIL-88 is not activated, or the sample is not MIL-88.

We are grateful to the reviewer's insightful comment, and indeed acknowledge that we have now found that the powder XRD pattern reported in our previous manuscript is associated with the formation of another polymorph other than MIL-88B. Thus, we have modified our method of synthesis to favour the crystallization of the MIL-88B polymorph. Moreover, we have simulated our data to that for MIL-88B from the literature, therefore verifying that the powder XRD patterns we now obtain indeed correspond to the MIL-88B diffraction profile. We have also analysed the MIL-88B crystal using TGA and BET and obtained results characteristic of MIL-88B.

We know that the samples fabricated in this method need to be washed to remove the byproducts, salts. Because the wash step is always, you can never really escape from re-activation steps.

The term 'activation' commonly refers to the process of removing unreacted solvents from the pores of the MOF structure. We have therefore clarified in the manuscript that in our work, this, i.e., the activation process, occurs *simultaneously* during synthesis.

While we agree that washing may typically be required for further applications, it does not alter the fact that our work constitutes the first time *simultaneous* activation during MOF synthesis has been claimed. We believe that this, together with the orientation of the crystals, comprises the importance of our work. Moreover, whether washing is included as part of the post-synthesis step or not is less important for our technique compared to other methods given that we do not use aggressive organic solvents such as DMF or methanol.

I could not understand the term "highly oriented" in the title. Every MOF single crystal is highly oriented, because every single crystal is oriented by definition of crystals.

We agree that every MOF *single crystal* is highly oriented. However, we wish to stress that we are not reporting single crystals but rather *powder* data. While powders in their bulk form synthesized through conventional means often suffer from poor orientation, the powders we synthesize using our reported method are freely detached from any surface and show preferred orientation.

In any case, the term 'oriented' is commonly used in the literature and we provide some references where the term is used. We have nevertheless removed the use of "highly" in keeping with the use of the term in the literature.

REVIEWERS' COMMENTS:

Reviewer #2 (Remarks to the Author):

The authors revised this manuscript thoroughly. I have no further comments on the technique part. But I have to point out one point which may be considered by the editors and the authors. Fig. 5 illustrated that the MIL-88 crystals might not be activated because of the existence of DMF. Activation of MOFs with large pores which are synthesized in high boiling point solvent is the most challenging part. MIL-101 may suffer more serious problem. Crystals with small pores or synthesized in low boiling-point solvents are usually easy to be activated or even be activated as-synthesized. The editors and authors may consider whether this new technique presented in this manuscript has really solved this problem.

Response to Reviewer #2's Comments

The authors revised this manuscript thoroughly. I have no further comments on the technique part. But I have to point out one point which may be considered by the editors and the authors. Fig. 5 illustrated that the MIL-88 crystals might not be activated because of the existence of DMF. Activation of MOFs with large pores which are synthesized in high boiling point solvent is the most challenging part. MIL-101 may suffer more serious problem. Crystals with small pores or synthesized in low boiling-point solvents are usually easy to be activated or even be activated as-synthesized. The editors and authors may consider whether this new technique presented in this manuscript has really solved this problem.

We are delighted to have addressed the reviewer's comments. We agree with the reviewer that the activation of MIL-88B and MIL-101 is quite challenging especially in high boiling point solvents. In Figure 5, the small BET surface area of the acoustically synthesized MIL-88B is attributed to the contraction of the pores upon the removal of the solvent during activation. This has been reported in the literature (Ma *et al. Cryst. Growth Des.* 2013, 13, 6, 2286-2291, Serre *et al. J. Am. Chem. Soc.* 124, 45, 13519-13526). Similarly, the removal of these solvents from large pore MOFs such as MIL-101 can be attained through changing the power supplied and the time of exposure of the MOF precursors to the acoustic field. Therefore, a brief discussion was added to the manuscript addressing this point as recommended by the reviewer.